# The Role of Vascularization in Nerve Regeneration: Mechanistic and Therapeutic Perspectives

**DOI:** 10.3390/ijms26178395

**Published:** 2025-08-29

**Authors:** Hamid Malekzadeh, Reade Otto-Moudry, Amy M. Moore

**Affiliations:** Department of Plastic and Reconstructive Surgery, The Ohio State University Medical Center, Columbus, OH 43210, USA; hamid.malekzadeh@osumc.edu (H.M.); reade.otto-moudry@osumc.edu (R.O.-M.)

**Keywords:** nerve injury, peripheral nerve, nerve regeneration, neural plasticity, vascularization

## Abstract

Peripheral nerve injuries (PNIs) are common and often result in sensorimotor deficits, chronic pain and decreased quality of life. While the peripheral nervous system has greater regenerative capacity than the central nervous system, recovery is often limited by intrinsic changes in the nerve and muscle. This review summarizes the process of nerve regeneration, with a focus on the role of the vasculature, following PNI and examines current bioengineering approaches to enhance peripheral nerve regeneration through modification of the nerve microenvironment and optimization of neurovascular interactions. The primary areas of translational research discussed in this review include vascularized nerve grafts, nerve conduits and scaffolds, bioactive peptides, nanoparticles, extracellular vesicles, stem cells, and gene therapy.

## 1. Introduction

Peripheral nerve injuries (PNIs) are common with an annual incidence of approximately 17 cases per 100,000 [1,2,3]. PNIs can result in sensorimotor deficits, neuropathic pain, and decreased quality of life [4,5]. In contrast to the central nervous system, the adult mammalian peripheral nervous system is capable of long-distance axonal regeneration and substantial functional recovery [6]. However, due to intrinsic changes in the nerve and muscle in prolonged denervation, functional recovery is often limited, creating a need for further research and understanding [6,7].

Extensive research has focused on identifying the intrinsic factors involved in axonal regeneration, such as cytoskeletal dynamics and transcriptional programs. It has become increasingly clear, however, that the microenvironment at the injury site—comprising immune cells, fibroblasts, microglia, and the microvasculature of the nerve—plays a key role in the regenerative capacity of damaged axons [8,9]. The peripheral nervous system’s enhanced regenerative capacity is largely attributed to its more supportive microenvironment and the presence of fewer inhibitory factors when compared to the central nervous system [10]. In addition to nervous system development, neurovascular interactions are key to nerve regeneration. The crosstalk between endothelial cells and other cells in the peripheral nervous system directs axonal regeneration and shapes the extracellular matrix (ECM) [11]. In this review we aim to summarize the basic science of the complex interactions between the nerve microenvironment and regenerating axons during the repair process after a PNI, with a focus on the instructive role of the vasculature and the cellular dynamic changes necessary for axonal regeneration. We will also review the translational research targeting vasculogenesis to improve peripheral nerve regeneration.

## 2. Methods

### 2.1. Literature Search

We searched the following terms in title/abstract: regeneration, nerve, neural, vascularization, neovascularization, and re-vascularization, along with relevant MeSH terms. We searched for studies in English listed in PubMed between January 1990 and November 2024. We also screened the references cited in the included articles and previous review articles. The search yielded 657 references. After screening titles and abstracts, 78 articles were included in this review.

### 2.2. Selection Criteria

We included basic science, translational, and clinical studies that investigated the basic science or therapeutic role of vascularization in axonal regeneration in peripheral nerves. Eligible studies assessed interventions aimed at enhancing vascularization and reported histological or functional outcomes related to nerve regeneration. Studies were excluded if the intervention’s primary goal was not to promote vascularization. Each reference was reviewed independently by two authors (HM and ROM), and any disputes regarding inclusion, exclusion, or data extraction were resolved by a third author (AMM).

## 3. Peripheral Nerve Injury and Regeneration

Peripheral nerves consist of bundles of axons and myelinating Schwann cells (SCs) embedded within connective tissue compartments. The larger axons are each myelinated by a single SC, which form segments of myelin separated by nodes of Ranvier [12]. The smaller axons are typically grouped in a ratio of approximately six axons per SC in structures called Remak bundles. A thin layer of vascular connective tissue, known as the endoneurium, surrounds individual myelinated axons or Remak bundles. These endoneurial fascicles are subsequently encased by the perineurium, a lamellated structure of fibroblast-like cells connected by tight junctions, forming part of the blood–nerve barrier. The outermost layer is the epineurium comprising fibroblasts, collagenous (type I) extracellular matrix, blood and lymphatic vessels, and tissue resident macrophages (Figure 1) [13,14].

Injury to a peripheral nerve triggers a complex multicellular response. Under physiological conditions, nicotinamide adenine dinucleotide (NAD^+^) is synthesized from the nicotinamide mononucleotide (NMN) by the enzyme nicotinamide mononucleotide adenylyltransferase 2 (NMNAT2). NMNAT2 is a labile protein that must be continuously transported from the neuronal soma into the axon. Following axonal injury, NMNAT2 transport is disrupted, and NAD+ is no longer synthesized in the distal nerve leading to an increase in the NMN/NAD^+^ ratio. In addition to facilitating the redox reactions necessary for ATP synthesis, NAD^+^ also allosterically inhibits the enzyme sterile alpha and TIR motif-containing protein 1 (SARM1). When NAD^+^ levels fall after injury, SARM1 is activated and further degrades NAD^+^, resulting in rapid depletion of cellular energy [15]. Following NAD+ depletion and loss of ATP, the subsequent influx of calcium activates calcium-dependent cysteine proteases, known as calpains. Calpains degrade cytoskeletal components, leading to axonal fragmentation and initiating the process of Wallerian degermation [16]. As axonal structures break down, resident cells secrete the chemokine CCL2, which facilitates macrophages migration into the injured nerve. These macrophages subsequently phagocytose myelin and axonal debris, secrete additional cytokines that promote a pro-regenerative milieu [17].

Additionally, retrograde changes occur in the proximal segment, potentially extending to the first or second node of Ranvier. Following axonal degeneration and the disruption of axon–SC interactions, denervated SCs undergo proliferation and dedifferentiate into progenitor-like cells with a regenerative phenotype. These dedifferentiated SCs organize into tubular cords called bands of Büngner and begin to migrate from the proximal stump [12]. Additionally, the neurons initiate a polarized form of regeneration, sprouting towards the distal stump [13,14].

Shortly after repair of a nerve transection, the two ends become reconnected by a newly formed tissue referred to as the bridge. The bridge initially has a non-directional structure consisting of fibroblasts, extracellular matrix, and inflammatory cells (mostly monocyte-derived macrophages) [14,18]. Neurites sprouting from the proximal stump must cross the bridge site to reach the distal stump and ultimately connect to their reinnervation target. Previous studies have shown that SCs facilitate this process by guiding the regenerating axons as they migrate across the bridge [19,20]. Parrinello et al. [20] demonstrated that SC cords migrate into the nerve bridge from both the proximal and distal stumps, with cords from the proximal stump guiding the regrowing axons. Their findings showed that the regenerating axons at the bridge site follow the SC cords’ trajectory, relying on interactions with SC processes for directional migration [18]. In addition to this structural role, SCs, along with infiltrating macrophages, secrete neurotrophic factors that promote axonal regrowth [11,21].

Fibroblasts are also involved in peripheral nerve regeneration [20,22]. After a nerve is cut, fibroblasts accumulate at the bridge site and secrete pro-inflammatory and pro-angiogenic cytokines. In addition, fibroblasts facilitate collective SCs migration across the bridge as compact cords by modifying SC behavior through Eph-ephrin signaling [20]. Eph-ephrin signaling is a mediator of cell positioning and can promote the directional movement of SCs by constraining migrating cells to specific areas through cell–cell adhesion or repulsion [23]. In a regenerating nerve, Eph-ephrin signaling alters the normal repulsive behavior of SCs causing them to favor adhesion and enables their collective movement. Specifically, EphB2 signaling induces expression of the transcription factor Sox2 in SCs. Sox2 promotes the relocalization of N-cadherin to cell–cell contact points, resulting in the aforementioned enhanced SC adhesion and facilitating cord formation [20]. Importantly, loss of Eph-ephrin signaling disrupts the formation of SC cords and their directional movement, resulting in aberrant axonal regrowth. These SC cords migrate collectively and guide directional growth of the axons across the bridge [20].

## 4. Role of Blood Vessels in Nerve Regeneration

Previous research by Cattin et al. demonstrated that blood vessels play a critical role in nerve regeneration by directing SC migration across the bridge toward the distal end of the nerve [14]. Initially, the bridge site lacks vascularization and becomes hypoxic. Macrophages at the bridge site detect this hypoxia and respond by releasing angiogenic factors such as vascular endothelial growth factor (VEGF). VEGF binds primarily to VEGF receptor 2 (VEGFR2), inducing receptor dimerization and autophosphorylation of intracellular tyrosine residues [24]. Phosphorylated VEGFR2 triggers three primary signaling pathways including the PI3K/Akt pathway, the MAPK/ERK pathway, and the PLCγ/PKC pathway, which lead to increased endothelial cell survival and proliferation, endothelial cell migration, and increased endothelial cell permeability and cytoskeletal rearrangement, respectively [25,26,27]. The subsequent stimulation of endothelial cell proliferation and migration results in migration of endothelial cells from the nerve ends into the bridge. Prior to SC migration, blood vessels extend from both the proximal and distal stumps into the bridge.

VEGF influences peripheral nerve repair beyond its angiogenic role by promoting and enhancing neurotrophic factor expression (NGF, BDNF) [28] It also modulates the immune response by increasing vascular permeability, facilitating macrophage infiltration, and promoting a regenerative, M2-polarized phenotype [29].

Cattin et al. showed that the alignment of blood vessels within the bridge corresponds with the direction of subsequent SC cord migration, suggesting that blood vessels provide guidance for SC movement [14]. Furthermore, SCs physically interact with the polarized blood vessels as they migrate along endothelial tubules. Notably, treatment of mice with Cabozantinib, a VEGF receptor inhibitor, following sciatic nerve transection and repair, inhibits both blood vessel entry into the bridge and the subsequent migration of SCs and axons [14] (Figure 2).

The alignment of neovessels within the bridge likely plays a critical role in effective scaffolding for SC migration. Injury-induced neovessels exhibit a dynamic phenotype marked by the temporary expression of Plexin-D1, a transmembrane receptor essential for vascular patterning and the orientation of endothelial cells within regenerating tissue. Plexin-D1, best known for its role in developmental vascular patterning, is specifically upregulated in angiogenic endothelial cell subtypes including tip cells, proliferating endothelial cells and immature endothelial cells. It acts to limit overactive VEGF signaling, helping to maintain appropriate vessel density and orientation and allowing for the formation of an organized endothelial scaffold across the bridge [11]. Early disruption of the Plexin-D1 pathway disturbs the alignment of neovessels in the bridge, resulting in long-term impacts on vascular organization and the orientation of the SC cords which typically follow these vascular pathways [11]. Overall, it has been demonstrated that VEGF-induced blood vessels guide SC and their accompanying axons during peripheral nerve regeneration (Figure 2).

Besides the role of vasculature as a scaffold for SCs, endothelial cells also promote nerve regeneration through the secretion and delivery of extracellular matrix proteins such as laminin and fibronectin, which are known to promote SC adhesion, migration, and axonal elongation. Additionally, endothelial cells secrete axonal guidance molecules including netrins, ephrins, and Slit proteins, which help coordinate the spatial orientation of blood vessels and regenerating axons [30,31]. Collectively, these molecular signals coordinate the crosstalk between the vasculature and elements of the regenerating nerve, creating a dynamic and instructive pro-regenerative microenvironment.

## 5. Therapeutic Approaches to PNI Repair

### 5.1. Nerve Graft Vascularization

Nerve grafts are often used when direct epineurial repair is not feasible due to gaps or excessive tension. A recent systematic review of numerous animal models showed that allografts often resulted in inferior outcomes compared to autografts [32]. Typically, sensory nerves are used in a reverse orientation to optimize the number of axons reaching their targets. Vascularization plays a significant role in autograft survival and regeneration, as insufficient vascularization may lead to central necrosis and create regions of necrosis and fibrosis that can obstruct nerve fibers and impair regeneration [33]. Re-vascularization of autografts occurs through inosculation, during which host tissue blood vessels infiltrate the graft and connect with its pre-existing avascular network [23]. Because of the importance of vascularization in the success of nerve repair and regrowth, there have been multiple approaches to surgically vascularize nerve autografts.

Vascularized nerve grafts involve the transfer of an autograft and its native blood supply into the defect (Figure 3). This allows neovascularization in the graft to begin before the onset of ischemia [7]. These grafts have been extensively evaluated in animal models [7,33,34,35,36,37,38,39,40,41]. A systematic review and meta-analysis reported that vascularized nerve grafts were associated with increased axon diameter, a higher number of regenerating axons, and improved nerve conduction velocity compared to non-vascularized autografts [42]. However, some studies have noted that the increased regeneration in vascularized grafts may plateau over long-term follow ups [7,34]. This may be attributed to variability in graft size across studies, where sufficient axonal regeneration eventually occurs in non-vascularized grafts as well, particularly in smaller defects. In addition, evidence from clinical case reports indicates the advantage of using vascularized nerve grafts becomes more apparent as the length of the nerve gap increases. In larger defects over 13 cm, vascularized grafts have been shown to support regeneration where non-vascularized autografts are often insufficient [43]. However, when applied to shorter gaps, vascularized grafts do not consistently outperform non-vascularized grafts [44].

However, vascularized nerve grafts are limited by having few appropriate donors and notable donor site morbidity. To avoid the donor site morbidity associated with harvesting a vascularized nerve graft, attempts have been made to prefabricate non-vascularized nerve grafts by implanting them between a vein and artery, or by implanting an arteriovenous fistula before harvesting the graft [45,46]. While there has been some evidence suggesting this approach may be clinically useful by reducing donor site morbidity, it requires an additional operative stage and delays definitive repair.

To improve the vascularity of nerve grafts, Lee et al. [47] delivered continuous VEGF to autografts using an osmotic pump, resulting in increased vascular density in treated rats by day three. However, by 16 weeks, no significant differences were observed between groups in compound muscle action potential (CMAP) or isometric tetanic force [48]. Given these mixed results and considering the technical challenges and potential complications of such interventions, there remains a need for alternative translational strategies to enhance nerve regeneration using grafts.

An alternative to auto- and isografts is the use of acellular nerve allografts. These do not require autologous tissue harvest and there is no associated donor site morbidity or sensory deficit [4]. Animal studies on nerve regeneration within acellular allografts indicate a gradual cellularization with endothelial and SCs. During the first week post-repair, these cells primarily accumulate near the distal ends of the graft, while the central region remains populated by extracellular matrix and other cell types such as macrophages [49]. By the second to third week, a vascular network forms within the regenerating tissue core, meeting the elevated metabolic needs for SC proliferation, myelination, and axonal growth. Although this vascular advancement occurs bidirectionally, growth predominantly progresses from the proximal end of the nerve [49,50].

There are currently several experimental approaches to improve outcomes by increasing vascularization in acellular allografts. In a rat model, surgical angiogenesis has been shown to improve proximal to distal vascularization of the allograft, potentially by contributing to a favorable immune environment [51,52]. Endothelial cells have also been used in conjunction with acellular allografts in a rat model to promote early angiogenesis, subsequently improving nerve regeneration and functional outcomes [53]. Additionally, acellular nerve grafts treated with nerve growth factor beta (β-NGF) and vascular endothelial growth factor (VEGF) have been shown to increase neovascularization, as well as the number and mean diameter of axons within the graft. However, these improvements plateaued at the 3- and 6-month follow-ups [54]. Another study using VEGF-treated acellular nerve grafts found similar axonal counts and areas at the distal coaptation site in both treated and untreated groups at 4 months. This may be because axonal regeneration eventually reaches comparable levels over time, diminishing the differences observed in earlier stages [48].

### 5.2. Vascularizing Nerve Conduits and Scaffolds

Nerve conduits, which act as a scaffold through which axons can grow across a small (usually <3 cm in humans) gap, can be composed of either biodegradable or non-biodegradable materials. Incorporating native vasculature, such as a subcutaneous artery, into silicone nerve conduits have been investigated as a method to enhance vascularization for nerve repair [53]. This technique, when used in a rat model, yielded an increased number of capillary structures within the conduit and an increase in intraneural microvasculature which is believed to have contributed to the observed differences in functional recovery [53]. While this technique demonstrated some promise, it was unable to improve outcomes in larger gaps [55]. Additionally, the use of non-biodegradable materials to form the nerve conduit may limit its current clinical application.

Recently, there have been multiple attempts to optimize both the biomaterials and constructs of nerve conduits to maximize vascularization without retaining non-biodegradable materials. Biomaterials such as fibronectin–laminin 1-laminin 2 (FibL1L2), perfluorotributylamine (PFTBA)/VEGF core–shell fibers, peptide nanofiber hydrogel, and fiber-reinforced 3D scaffolds containing endothelial cells and SCs have all shown improved vascularization in various animal models. In a rat model with a 15 mm sciatic nerve gap, FibL1L2 conduits showed increased angiogenesis and enhanced axonal density compared to unmodified conduits. The functional outcomes of FibL1L2 conduits were comparable to those of autografts. The success of FibL1L2 conduits is discussed to be largely predicated on the impact of laminin-1 on angiogenesis and the inflammatory profile of fibronectin [56]. Perfluorotributylamine/VEGF core–shell fiber conduits rely on the oxygen carrying capacity of perfluorotributylamine to combat hypoxia in the initially non-vascularized bridge tissue and later on VEGF to promote angiogenesis [57]. Like FibL1L2 conduits, PFTBA/VEGF core–shell fibers promoted angiogenesis and led to improved axonal regeneration, sciatic function index, and muscle weight ratio in 15 mm nerve gaps in rats, with functional outcomes comparable to autografts. Self-assembling nanofiber hydrogels functionalized with VEGF- and brain-derived neurotrophic factor (BDNF)-mimetic peptides have also been shown to increase neovascularization as well as the density of regenerating axons likely as a result of the known impact of VEGF on angiogenesis and the promotion of endothelial cell survival and vessel stability by BDNF [58].

Unlike approaches relying on the properties of biomaterials or growth factors, another study engineered a pre-vascularized tissue conduit by culturing fiber-reinforced 3D scaffolds with endothelial cells, and SCs showed increased axonal regeneration, myelination, and functional recovery in a rabbit sciatic nerve injury model [59]. Although vascularized nerve conduits have promising outcomes, they have yet to be clinically translated. Major barriers include variability in biocompatibility, risks of thrombosis or infection, and limited long-term safety and efficacy data in human studies [60].

### 5.3. Bioactive Peptides

Numerous bioactive peptides—protein components that become active upon cleavage—exhibit well-documented angiogenic properties [61]. As discussed in previous sections, VEGF has been shown in numerous studies to increase angiogenesis, axonal outgrowth, and subsequent SC migration. In addition to VEGF, BDNF has also been studied in conjunction with nerve conduits and shown to improve myelination and axon outgrowth [62] (Figure 3).

Laminin, a bioactive peptide derived from the extracellular matrix, has been studied for its role in angiogenesis. It is believed that laminin-1 promotes the differentiation of endothelial cells through regulation of gene and protein expression [63]. In the peripheral nervous system, laminin is essential for axonal myelination and regeneration [31]. In a rat model with an 18 mm nerve gap, a fibrin–laminin serum significantly enhanced the extent of mature nerve regeneration reaching the target muscle [64].

Despite their promise, bioactive peptides are limited by poor stability, rapid enzymatic degradation, high cost, and short in vivo half-lives, all of which reduce their therapeutic efficacy [65]. Sustained, localized delivery remains challenging due to rapid clearance, proteolytic breakdown, and poor tissue retention, while systemic administration may lead to off-target effects or immune responses [66].

### 5.4. The Role of Nanoparticles in Vascularization and Repair

Nanoparticles (1–100 nm) have been increasingly investigated as therapeutic adjuncts in the management of peripheral nerve injury. One key application involves their use as carriers for bioactive molecules, as they address challenges associated with bioactive peptides, including rapid enzymatic degradation, poor tissue retention, and the need for sustained, localized delivery. In a rat sciatic nerve injury model, delivery of nitric oxide-releasing silica nanoparticles was shown to enhance re-vascularization and support nerve regeneration [47]. In addition to their delivery capabilities, nanoparticles have also been explored for their magnetic properties to further promote angiogenesis and functional recovery. A commercially available conduit injected with polyethylenimine (PEI)-coated iron oxide nanoparticles functionalized with VEGF and NGF significantly enhanced nerve regeneration and motor recovery compared to a conduit injected with free NGF and VEGF [67]. Despite these promising results, the storage and controlled release of nanoparticles remain significant challenges.

### 5.5. Extracellular Vesicles

Extracellular vesicles (EVs) are lipid-bound vesicles secreted by cells that play key roles in intercellular communication by delivering proteins, DNA, and RNA [68]. EVs derived from various stem cell types have been studied in PNI models for their regenerative and angiogenic potential. Mesenchymal stem cell (MSC)-derived EVs, in particular, are attractive due to their availability and ease of manufacturing, and have been shown to enhance axonal outgrowth and functional recovery in rat models [69]. Similarly, EVs from adipose-derived stem cells (ADSCs) improved nerve regeneration in a rat sciatic nerve crush model, likely by promoting Schwann cell activity [70].

Endothelial cell-derived EVs have also been shown to enhance vascularization and improve functional outcomes, including sciatic function index and CMAP recovery [71]. In another study, EVs from human umbilical vein endothelial cells (HUVECs) transfected with Netrin-1 promoted angiogenesis, axon regeneration, and myelination, along with improved sciatic function index, CMAP, and paw withdrawal threshold in a rat sciatic nerve crush model [72]. Pericyte-derived nanovesicles enhanced early-stage angiogenesis and nerve regeneration in a mouse sciatic nerve transection and repair model, with improved motor and sensory function observed after 8 weeks of treatment [73]. Furthermore, Schwann cell-derived exosomes also supported intraneural re-vascularization following nerve injury, and their angiogenic potential was further enhanced under hypoxic conditions [74].

The clinical translation of EV-based therapies is limited by several challenges. EV biodistribution and short in vivo circulation (due to rapid clearance and liver/spleen accumulation) raise concerns over the inability to maintain therapeutic concentrations at injury sites and off-target effects. EVs are heterogeneous, varying in size, cargo, and function even when derived from clonal cell populations. Additionally, conventional isolation methods (e.g., ultracentrifugation, precipitation) often yield products of variable purity, leading to batch-to-batch variability and contamination with non-EV components [75]. This necessitates the need for robust good manufacturing practice (GMP) protocols before translation to clinical-grade EVs [76].

### 5.6. Stem Cells

The therapeutic use of stem cells in PNI relies on their ability to enhance angiogenic and neurotrophic factors, differentiate into endothelial and Schwann-like cells, and promote myelination [77]. Among MSCs, ADSCs have been studied most extensively due to their ease of harvest, high yield, and simpler preparation [78]. ADSCs support angiogenesis through both differentiation into endothelial cells and the secretion of neurotrophic factors such as NGF, BDNF, and VEGF [79]. Because of these properties, ADSCs have been used to augment acellular nerve allografts in rodent sciatic nerve injury models. Acellular nerve allografts seeded with ADSCs have demonstrated increased vascular volume and surface area [52,80]. Additionally, decellularized allografts wrapped with ADSC sheet achieved outcomes comparable to autografts in terms of fascicular cross-sectional area and CMAP [81].

Other MSCs studied in the context of PNI include bone marrow MSCs and multipotent vascular stem cells (MVSCs) (Figure 3). Implantation of bone marrow MSCs into a nerve conduit containing a vascular bundle promoted CMAP amplitude; axonal area and myelin thickness; and the tibialis anterior muscle weight [82,83]. Bone marrow MSCs have been studied in conjunction with acellular nerve allografts and demonstrated increased angiogenesis and nerve regeneration. It was also observed that some of the bone marrow MSCs differentiated into Schwann-like cells, which is a possible explanation for the observed improvement in outcomes [83,84]. MVSCs have been studied in nerve conduits, and were found to improve CMAP, microvessel formation, SC migration, and axon growth and myelination [85]. Despite promising outcomes, these approaches have not yet been clinically translated.

Most pre-clinical in vivo stem cell therapy studies showed promising results, there were only six cases reported on stem cell therapy for PNI [86,87], indicating that enough clinical evidence is not yet available to allow their clinical implementation. Although most clinical studies in other fields reported that MSCs are relatively safe and effective, their clinical use is limited by considerable variability in cell properties due to differences in donor age, tissue source, and processing techniques, undermining reproducibility and potency assessment [88]. Furthermore, like many biologics-based drugs, current ADSC isolation and culture protocols lack standardization, resulting in inconsistent cellular yield, secretome profile, and therapeutic efficacy; regulatory guidelines for dosing, safety monitoring, long-term follow-up, and standardized GMP-compliant scale-up are still needed.

### 5.7. Gene Therapy

Gene therapy has gained significant attention in recent years as a potential treatment approach for PNIs. VEGF gene therapy, in particular, has been studied using both viral and non-viral transfection methods, and has demonstrated significant potential in promoting neuronal survival, angiogenesis, and peripheral nerve regeneration (Figure 3). One study demonstrated that VEGF gene therapy delivered via injection of viral vectors around nerve autografts in a rat model led to a higher fraction of large axons, and enhanced sciatic function index [89].

Although viral transfection offers high efficiency and stable gene expression, its use is limited by safety concerns, such as immune responses and the risk of insertional mutagenesis, prompting interest in non-viral methods which can be safer and more versatile [90]. Other studies have investigated non-viral methods of gene delivery. Pereira Lopes et al. used electroporation to introduce VEGF and granulocyte colony-stimulating factor genes into the sciatic nerve and adjacent muscle in a mouse model, which was transected and placed in an empty polycaprolactone nerve guide. They showed an increase in myelination and angiogenesis, as well as earlier and more robust functional recovery [91]. Another study used a similar method of local injection and electroporation to introduce VEGF genes into an autologous sciatic nerve graft in mice. This study demonstrated an increased number of myelinated axons, increased angiogenesis, increased gastrocnemius weight, and an improved sciatic function index [92]. These studies indicate that gene therapy aimed at enhancing angiogenesis after peripheral nerve injury may be a promising avenue; however, they have yet to be clinically translated.

## 6. Conclusions

Despite the peripheral nervous system’s capacity for axonal regeneration, functional recovery is often limited, particularly in cases of large defects, complex injuries, or prolonged denervation. Vascularization, as discussed above, is a critical component of nerve healing. Endothelial cells migrate bidirectionally across the bridge and guide the migration of SC cords, which ultimately guide axon regrowth. Vascularization also serves to deliver essential oxygen and nutrients, as well as deliver neurotrophic factors to the site of injury.

Given the importance of vascularization for nerve regeneration, many recent advancements, including direct vascularization, vascularized or vasculogenic conduits and scaffolds, bioactive peptides, nanoparticles, extracellular vesicles, stem cells, and gene therapy, focus on enhancing re-vascularization. While promising, there are many approaches to the management of PNIs that have yet to be clinically translated, and additional avenues, such as tissue and cell reprogramming, that have yet to be fully explored. It is our hope that this review can help to summarize and organize the current basic sciences research and promote the continuation of this essential research, especially as it relates to the impact of vascularization on axonal regrowth and peripheral nerve regeneration.

## Figures and Tables

**Figure 1 ijms-26-08395-f001:**
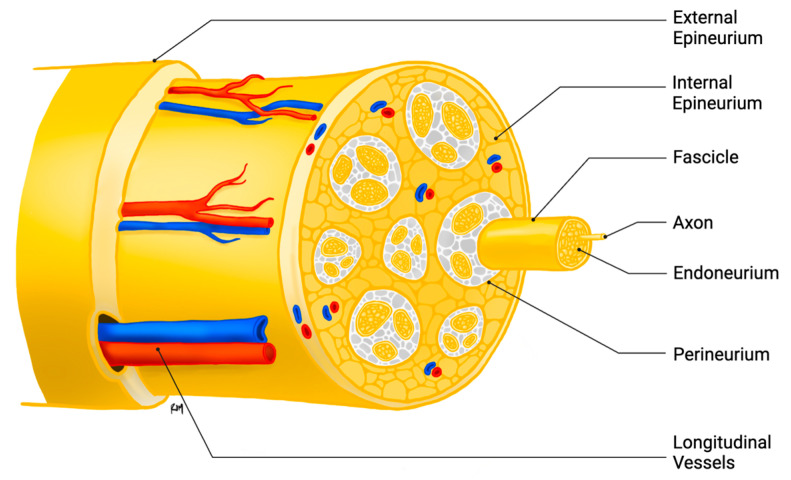
Structure of the peripheral nerve.

**Figure 2 ijms-26-08395-f002:**
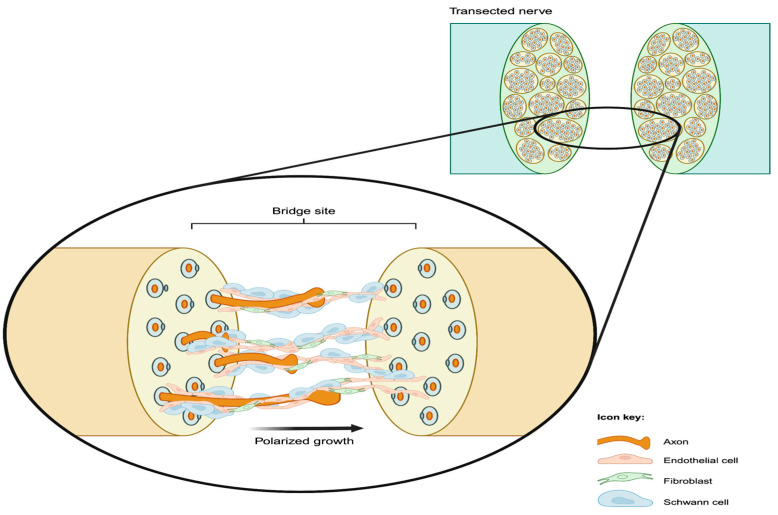
Illustration of the bridge formed across a transected nerve by bands of endothelial cells, followed by the cooperative polarized growth of Schwann cells and fibroblasts, which compose the bands of Brügner and ultimately guide the advancement of axons from the proximal stump. Image created in https://BioRender.com.

**Figure 3 ijms-26-08395-f003:**
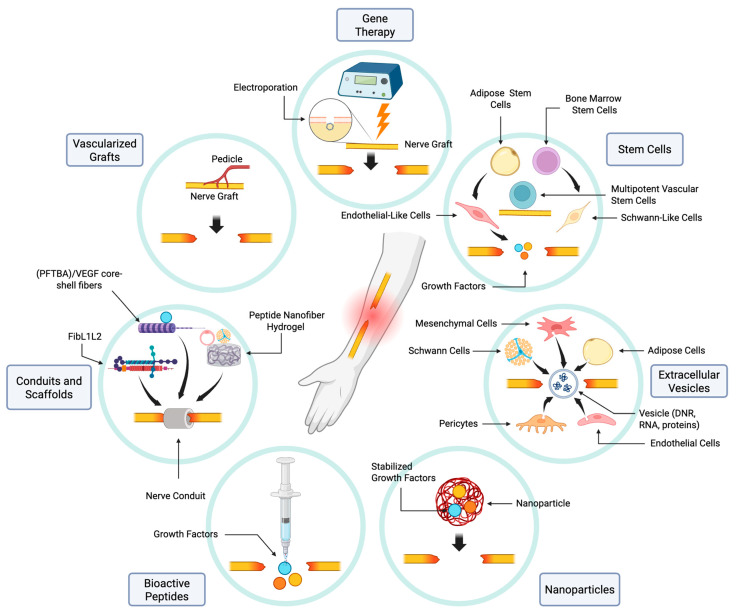
Bioengineering approaches to improving nerve regeneration following PNI include vascularizing nerve grafts, conduits and scaffolds composed of materials such as FibL1L2, (PFTBA)/VEGF core–shell fibers, peptide nanofiber hydrogels, the use of bioactive peptides such as VEGF and BDNF, nanoparticles which stabilize various growth factors for prolonged delivery to the injured nerve, extracellular vesicles derived from various cell lines and containing DNR, RNA or various proteins, stem cells derived from adipose, bone marrow, and multipotent vascular cell lines, and gene therapy in order to improve vascularity. Image created in https://BioRender.com.

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
