# Peer review of "The Role of Vascularization in Nerve Regeneration: Mechanistic and Therapeutic Perspectives"

_ijms, 2025, doi:10.3390/ijms26178395_

Round 1

Reviewer 1 Report

Comments and Suggestions for Authors

The manuscript presents a concise review of current approaches for increasing vascularisation of peripheral nerve grafts. It briefly describes cellular processes during nerve regeneration through tubular grafts and extends towards recent findings with special focus on vascular support. The text is informative, up to date, and could deliver impulses for future research.

Major points:

1) Legend Figure 2: The legend should explain also the role of the depicted fibroblasts.

2) Legend Figure 3: The symbols for cell types and material properties need to be explained in more detail

Minor points:

1) references in lines 81, 281, 287, 303, 332: check for type setting.

2) Line 146: check for word doubling

Author Response

Major points:

Comment 1) Legend Figure 2: The legend should explain also the role of the depicted fibroblasts.

- The legend for Figure 2 has been updated to more clearly explain the roles of different cell types and elucidate the mechanism of peripheral nerve regrowth following transection.

Comment 2) Legend Figure 3: The symbols for cell types and material properties need to be explained in more detail

- Figure three has been updated to better reflect the specific molecular mechanisms discussed in the paper and has been labeled more clearly in order to better explain the depicted materials and mechanisms. The legend for Figure 3 has been updated to reflect these changes.

Minor points:

Comment 1) references in lines 81, 281, 287, 303, 332: check for type setting.

- corrected

Comment 2) Line 146: check for word doubling

- corrected

Reviewer 2 Report

Comments and Suggestions for Authors

Although the topic is highly relevant and timely, the manuscript lacks the depth, critical synthesis, and originality expected in a state-of-the-art review. The content is primarily descriptive, offering superficial summaries of numerous studies without adequately discussing their limitations, contextual relevance, or translational challenges.

Moreover, the review fails to:

  • Present a clear methodology for literature selection. There is no description of how the authors selected the literature cited. A rigorous review must define: Inclusion/exclusion criteria for studies (e.g., publication date, model species, study type). Databases used (e.g., PubMed, Scopus). Search strategy and keywords. Criteria for evaluating study quality or translational relevance. Include a Methods section outlining the review protocol. If this is a narrative review, justify the lack of systematic methods and clearly define the scope.
  • Offer a critical comparison between studies, especially where contradictory results exist.
  • Identify knowledge gaps or propose future research directions with any meaningful insight.
  • Discuss in sufficient detail the translational barriers and safety concerns associated with advanced therapeutic strategies such as gene therapy, nanoparticles, and extracellular vesicles.

The figures, while visually organized, contribute little beyond basic illustration and are not fully integrated with the text. In particular, Figure 2 oversimplifies complex processes and lacks adequate mechanistic detail, while Figure 3 is overcrowded and lacks conceptual clarity.

Author Response

Comment 1) Present a clear methodology for literature selection. There is no description of how the authors selected the literature cited. RA rigorous review must define: Inclusion/exclusion criteria for studies (e.g., publication date, model species, study type). Databases used (e.g., PubMed, Scopus). Search strategy and keywords. Criteria for evaluating study quality or translational relevance. Include a Methods section outlining the review protocol. If this is a narrative review, justify the lack of systematic methods and clearly define the scope.

Response to comment 1) We have now included a Methods section that describes our literature search and selection process. To identify relevant articles, we used the following PubMed search query: ("Nerve Regeneration"[MeSH Terms] OR "regenerat*"[Title/Abstract]) AND ("Peripheral Nerves"[MeSH Terms] OR "nerve"[Title/Abstract] OR "neural"[Title/Abstract]) AND ("vascularization"[Title/Abstract] OR "neovascularization"[Title/Abstract] OR "revascularization"[Title/Abstract] OR ("neovascularization, physiologic"[MeSH Terms] OR "Angiogenesis"[MeSH Terms]))

For clarity and flow, we referenced key search terms in the main text rather than including the full query. The Methods section also outlines our inclusion criteria and screening process. Since this is a narrative review, we focused on selecting high-impact, relevant studies with translational significance, and we clarified this scope in the revised manuscript.

Comment 2) Offer a critical comparison between studies, especially where contradictory results exist.

Response to comment 2) We have added some explanations for contradictory findings among included articles. These additions aim to highlight differences in methodology, animal models used, and interpretations of results.

Comment 3) Identify knowledge gaps or propose future research directions with any meaningful insight. Discuss in sufficient detail the translational barriers and safety concerns associated with advanced therapeutic strategies such as gene therapy, nanoparticles, and extracellular vesicles.

Response to comment 3) We have added a subsection at the end of each relevant section discussing limitations on clinical translation and possible directions. These additions are highlighted in yellow in the manuscript.

Comment 4) The figures, while visually organized, contribute little beyond basic illustration and are not fully integrated with the text. In particular, Figure 2 oversimplifies complex processes and lacks adequate mechanistic detail, while Figure 3 is overcrowded and lacks conceptual clarity.

Response to comment 4) We revised the captions for both figures to improve clarity and better integrate them with the main text. Additionally, we made changes to Figure 3 for clarity.

Round 2

Reviewer 2 Report

Comments and Suggestions for Authors

The scope remains broad, and the review sometimes reads as a narrative listing of studies rather than a focused attempt to answer a specific mechanistic or translational question. A more explicit statement of the review’s central hypothesis (e.g., that targeting vascularization could meaningfully accelerate translation to clinical practice) would strengthen the framing.

The manuscript partially fills a gap by compiling therapeutic strategies, but it still lacks a sufficiently critical appraisal of why previous translational efforts have failed and what specific barriers remain.

Limited critical comparison between studies with conflicting outcomes.

Insufficient mechanistic synthesis, e.g. VEGF is discussed mainly descriptively without integrating how its effects interact with Schwann cell biology or inflammatory modulation.

Future research directions are mentioned but lack specificity or prioritization (e.g., which preclinical models or delivery strategies are most promising for translation?).

No justification for restricting the search to PubMed; inclusion of Embase or Scopus would improve comprehensiveness.

The process for evaluating study quality remains vague. A narrative review can be justified, but clearer criteria for prioritizing evidence (e.g., weighting in vivo over in vitro data) are needed.

Provide a clearer synthesis of which therapeutic approaches have the strongest preclinical evidence versus those still at a conceptual stage.

Author Response

 To the reviewers, thank you so much for the time and energy you’ve invested in reading and thinking about this manuscript. We appreciate your feedback and have made the below changes in response to your suggestions:

1)  The scope remains broad, and the review sometimes reads as a narrative listing of studies rather than a focused attempt to answer a specific mechanistic or translational question. A more explicit statement of the review’s central hypothesis (e.g., that targeting vascularization could meaningfully accelerate translation to clinical practice) would strengthen the framing.

  • We aimed to write an article to give an overview or summary of the current research as it related to the role of vasculogenesis in peripheral nerve regeneration and hope we have addressed the framing by explaining the importance of neovascularization for nerve regeneration and the statement of purpose at the end of the introduction.

2)  The manuscript partially fills a gap by compiling therapeutic strategies, but it still lacks a sufficiently critical appraisal of why previous translational efforts have failed and what specific barriers remain

  •  We hope to have addressed the barriers that remain to translation in individual sections (lines 237-239, 274-277, 291-295, 308-309, 330-339, 374-383). Most of the studies were in small animals and in most cases they have reported promising results but larger animal studies are usually required before translation into humans.

3) Limited critical comparison between studies with conflicting outcomes.

  • We hope that our discussion about the likely causes of variability between outcomes will address this question. Unfortunately, variability in injury models and the methodologies as well as reported outcomes, making them poor candidates for direct comparison. (lines 237-239, 330-339, 376-383)

4) Insufficient mechanistic synthesis, e.g. VEGF is discussed mainly descriptively without integrating how its effects interact with Schwann cell biology or inflammatory modulation

  • We agree that the mechanistic details of nerve regeneration discussed in this review do not represent the entire picture. Our intention was to focus on the basic science aspects of vasculogenesis so the reader can link these basics to the therapeutic approaches that was discussed in the second part of the paper. We added a brief description of the role of VEGF on Schwann cell biology or inflammatory modulation.
  • To maintain concision, we consolidated the mechanistic discussion at the beginning of the paper rather than repeating it in each section, as many mechanisms (e.g., those involving VEGF) are common across study modalities. Furthermore, the specific mechanisms proposed by individual research groups are often not fully elucidated in their publications, so we have been cautious in assigning mechanistic hypotheses.

5)  Future research directions are mentioned but lack specificity or prioritization (e.g., which preclinical models or delivery strategies are most promising for translation?)

  • While we believe many of these avenues may be promising, and hope to see future research being pursued in all of the categories we discussed, we refrained to prioritize one methodology over another given the extreme heterogeneity of the research methodologies, as well as the various logistical barriers to clinical translation discussed.

6)     No justification for restricting the search to PubMed; inclusion of Embase or Scopus would improve comprehensiveness.

  • This topic is very broad, and while we wanted to provide a comprehensive overview of the subject, we also wanted to constrain our scope somewhat. We chose PubMed because it is the dominant search platform for clinical and biomedical research, and we believed it will be comprehensive enough for our goal.

7)  The process for evaluating study quality remains vague. A narrative review can be justified, but clearer criteria for prioritizing evidence (e.g., weighting in vivo over in vitro data) are needed.

  • Our goal was to provide a broad overview of the basic science and therapeutic strategies related to vasculogenesis in peripheral nerve regeneration rather than to address a specific research question.  the inclusion criteria were intentionally broad, encompassing studies with varied outcome measures, which precludes drawing definitive conclusions.

8)  Provide a clearer synthesis of which therapeutic approaches have the strongest preclinical evidence versus those still at a conceptual stage.

  • Same as above

Thank you again for your feedback, and we hope we were able to address the majority of. Your concerns and suggestions in our revised manuscript.

Round 3

Reviewer 2 Report

Comments and Suggestions for Authors

I have no comments